# Seed- and Soil-Dependent Differences in Murine Breast Tumor Microenvironments Dictate Anti-PD-L1 IgG Delivery and Therapeutic Efficacy

**DOI:** 10.3390/pharmaceutics13040530

**Published:** 2021-04-10

**Authors:** Yan Ting Liu, Shreya Goel, Megumi Kai, Jose Alberto Moran Guerrero, Thao Nguyen, Junhua Mai, Haifa Shen, Arturas Ziemys, Kenji Yokoi

**Affiliations:** Houston Methodist Research Institute, Houston, TX 77030, USA; yliu4@houstonmethodist.org (Y.T.L.); sgoel@mdanderson.org (S.G.); MKai@mdanderson.org (M.K.); A01153856@itesm.mx (J.A.M.G.); tnguyen1d626@houstonmethodist.org (T.N.); jmai@houstonmethodist.org (J.M.); HShen@houstonmethodist.org (H.S.); aziemys@houstonmethodist.org (A.Z.)

**Keywords:** immunotherapy, heterogeneity, drug delivery, tumor microenvironment

## Abstract

We sought to determine if Stephen Paget’s “seed and soil” hypothesis of organ-preference patterns of cancer metastasis can explain the development of heterogeneity in a tumor microenvironment (TME) as well as immunotherapeutic delivery and efficacy. We established single-cell-derived clones (clones 1 and 16) from parental 4T1 murine breast cancer cells to create orthotopic primary and liver metastasis models to deconvolute polyclonal complexity cancer cells and the difference in TME-derived heterogeneities. Tumor-bearing mice were treated with anti-PD-L1 IgG or a control antibody, and immunofluorescent imaging and quantification were then performed to evaluate the therapeutic efficacy on tumor growth, the delivery of therapy to tumors, the development of blood vessels, the expression of PD-L1, the accumulation of immune cells, and the amount of coagulation inside tumors. The quantification showed an inverse correlation between the amount of delivered therapy and therapeutic efficacy in parental-cell-derived tumors. In contrast, tumors originating from clone 16 cells accumulated a significantly greater amount of therapy and responded better than clone-1-derived tumors. This difference was greater when tumors grew in the liver than the primary site. A similar trend was found in PD-L1 expression and immune cell accumulation. However, the change in the number of blood vessels was not significant. In addition, the amount of coagulation was more abundant in clone-1-derived tumors when compared to others. Thus, our findings reconfirmed the seed- and soil-dependent differences in PD-L1 expression, therapeutic delivery, immune cell accumulation, and tumor coagulation, which can constitute a heterogeneous delivery and response of immunotherapy in polyclonal tumors growing in different organs.

## 1. Introduction

Despite significant advancements in the diagnosis and surgical therapies for patients with primary tumors, the majority of deaths from cancer are caused by the metastases of tumors [1,2]. Heterogeneity in metastatic tumors’ response to systemic anticancer therapies, including immune checkpoint inhibitors (ICI), has significantly limited clinical benefits for breast cancer patients [3,4]. Understanding the specific driving forces behind these heterogeneities will facilitate greater insight into metastatic breast cancer’s therapeutic resistance.

Extensive preclinical and clinical research has confirmed Stephen Paget’s original “seed and soil” hypothesis that proposed that the organ-preference patterns of cancer metastasis are the product of interactions between metastatic cancer cells (the “seed”) and their organ microenvironment (the “soil”) [5,6]. Studies have also revealed that tumors in patients and established cancer cell lines are composed of multiple genetically distinct clones with different phenotypes [7,8]. Polyclonal cancer cell (seed) crosstalk with the tumor microenvironment (TME) at different organ sites (soil) can further increase the complexity of heterogeneities.

These spatial heterogeneities in cancer cells and the TME can affect the delivery of systemically injected chemotherapeutics to tumors and therapeutic efficacies [9,10,11]. The delivery of therapeutics across biological barriers, including tumor-associated blood vessels and stroma, can significantly affect the efficacy of cancer therapies [9,12]. Insufficient delivery of chemotherapeutics below the threshold of concentrations that induce biological effects on cancer cells can generate “drug delivery-based therapeutic resistance” [13,14]. Nevertheless, the delivery of ICI to tumor cells and its correlation with therapeutic efficacy have not been fully investigated. In the current study, we sought to determine that the heterogeneous response of tumors to ICI can be associated with seed- and soil-dependent differences in the amount of drug delivery. For this purpose, we used single-cell-derived clones from parental 4T1 murine breast cancer cells to create orthotopic primary and liver metastases models in syngeneic mice to deconvolute the complexity of polyclonal cancer cells and different TME-derived heterogeneities. We utilized intravital microscopy for the image delivery of systemically injected fluorescently labeled anti-PD-L1 IgG to an individual tumor and its therapeutic response through a window chamber for seven days. Phenotypic in vivo therapeutic study and imaging analysis of tumors established from single-cell-derived clones growing in different organs may elucidate the specific driving forces in creating a heterogeneous response to anti-PD-L1 IgG therapy.

## 2. Materials and Methods

### 2.1. Cell Culture

The 4T1-luc2-td tomato murine breast cancer cell line was obtained from Perkin Elmer (Waltham, MA). The cells were cultured in a complete minimal essential medium supplemented with 10% fetal bovine serum (Life Technologies, Inc., Grand Island, NY, USA) and a penicillin–streptomycin cocktail (Flow Laboratories, Rockville, MD, USA). Single-cell-derived clones were established from the 4T1-luc2-td parental cells by using a serial dilution method until a single-cell per 200 µL of the medium was obtained. A single-cell in the medium was then seeded into a 96-well plate to grow and expand in a CO_2_ incubator at 37 °C to establish clonal cell lines. Among other established clones, we selected clone 1 and clone 16 cells for our current study.

### 2.2. Clonogenic Assay

Five hundred cells (parental, clone 1, and clone 16 cells) were plated in 6-well plates (Thermo Fisher Scientific, Waltham, MA, USA) with fresh media. The plated cells were incubated in a CO_2_ incubator at 37 °C for 3 days, and then the cells were fixed with acetic acid in methanol in 1 to 7 dilution. Lastly, a 0.5% crystal violet solution (Sigma, Burlington, MA, USA) was added to the wells and incubated at room temperature for 2 h. An EVOS microscope (Life Technology, Carlsbad, CA, USA) was used to capture images for analysis.

### 2.3. Angiogenesis Array

Cell culture supernates of parental, clone 1, and clone 16 cells were obtained and analyzed for protein expression with the use of a mouse angiogenesis array kit (R&D Systems, Minneapolis, MN, USA) by following the manufacturer’s instructions. Blots were analyzed with ImageJ software (NIH, Bethesda, MD, USA) [15].

### 2.4. Mice

Female BALB/c mice, 6–8 weeks of age, were purchased from Charles River Laboratories (Wilmington, MA, USA). The mice were maintained in animal facilities at the Houston Methodist Research Institute approved by the American Association for Accreditation of Laboratory Animals.

### 2.5. Establishment of Primary Tumor and Experimental Liver Metastases

4T1 parental, clone 1, or clone 16 cells (1 × 10^5^/100 µL) were injected into the mouse mammary fat pad (mfp) to create three sets of primary tumors. To establish experimental liver metastases, 4T1 parental, clone 1, or clone 16 cells (1 × 10^5^/100 µL) were injected into the mice’s spleens to create three metastases models followed by splenectomy. The cells injected into the spleen disseminated to the liver through the portal vein, producing multiple liver metastases [14,16]. All the surgical procedures in this study were approved by the Institutional Animal Care and Use Committee (IACUC) of the Houston Methodist Research Institute.

### 2.6. Therapy

Seven days after the inoculation of tumor cells, mice bearing primary tumors or liver metastases were randomly picked from the cages to receive intravenous (iv) injection of either rat isotype control IgG (15 mg/kg) or rat anti-PD-L1 IgG (15 mg/kg) on days 7 and 10 (both from Bio X Cell, Lebanon, NH, USA). The length and width of the primary tumor were measured every other day from the beginning of days 5 to 21, and tumor volume (*V*) was calculated using the formula *V* = ½ (length × width^2^).

### 2.7. Intravital Microscopy Imaging

Mouse-bearing liver metastases were implanted with a window chamber in the abdominal wall for IVM imaging (Nikon Inc., Tokyo, Japan) [16,17]. Anti-PD-L1 IgG or isotype control IgG (BioLegend, San Diego, CA, USA) conjugated with brilliant violet 421 (200 µg/kg) were iv injected into the tumor-bearing mice 7 days after the tumor cell inoculation. The delivery of fluorescently labeled anti-PD-L1 IgG to liver metastases was imaged at 0 and 6 h after the injection. Tumor-bearing mice were then treated with either isotype control Ab or anti-PD-L1 IgG (15 mg/kg) on days 7 and 10, as described above. Therapeutic efficacy was monitored by imaging the tumor diameter every day for 7 days through a window chamber of anesthetized mice. The images were processed to assess the fluorescent intensity of anti-PD-L1 IgG inside the tumor and to measure the tumor size at different time points using NIS-element image processing software (Nikon Inc., Tokyo, Japan) [16,17].

### 2.8. Immunofluorescent Imaging

Frozen sections of the primary tumor and liver metastases were fixed with 4% paraformaldehyde in PBS and blocked with blocking solution (5% horse sera and 1% goat sera in PBS). Immunohistochemical staining of tissue sections was performed using antibodies against PD-L1 (BioLegend, San Diego, CA, USA), CD31 (BD Biosciences, San Jose, CA, USA), fibrinogen (Abcam, Cambridge, MA, USA), and various immune cells, such as CD8 (Invitrogen, Waltham, MA, USA), F4/80 (Abd Serotec, Raleigh, NC, USA), and CD45 (BD Pharmingen, San Diego, CA, USA). To image delivery of iv injected rat anti-PD-L1 IgG or isotype control IgG in tumors, the tissue sections were incubated with Alexa-fluor 647-labeled anti-rat IgG (Jackson ImmunoResearch, West Grove, PA, USA). All images were acquired using laser scan confocal microscopy (Nikon Inc., Tokyo, Japan). ImageJ software (NIH, Bethesda, MD, USA) was used to calculate the area fractions of positive staining on the captured fluorescence images [15]. All graphs are plotted based on the area fractions, and each symbol represents an individual imaging analysis.

### 2.9. Statistical Analysis

Results are expressed as the mean ± SD, and differences between groups were assessed by the Mann–Whitney test using GraphPad Prism, version 8.3.0 (GraphPad Software, San Diego, CA, USA). All *p* values less than 0.05 were considered statistically significant.

## 3. Results

### 3.1. Establishment of Single Clonal Cell from 4T1 Parental Cells

We established single-cell-derived clonal cell lines to deconvolute the complexity of polyclonal (parental) 4T1 murine breast cancer cells. Among several established clonal cell lines, we selected clones 1 and 16 to perform experiments in addition to the parental cells. A clonogenic assay was performed to compare colony phenotypes among cell lines and within the same cell line. While parental cells formed heterogeneous colonies in terms of the density and shape of the cells, clone 1 and clone 16 produced clone-dependent homogeneous colonies (Figure 1). In addition, we performed in vitro studies for the parental and clone cells to further investigate the difference in angiogenesis protein production. However, the data from the angiogenesis array did not clarify the heterogeneous response to anti-PD-L1 IgG therapy (Appendix A).

### 3.2. Determine the Seed- and Soil-Dependent Differences in Therapeutic Efficacy of Anti-PD-L1 IgG

To determine whether there is a seed- (clone) and soil-dependent (TME) difference in the efficacy of immunotherapeutic, we treated mice bearing both a primary tumor and liver metastases with either isotype control or anti-PD-L1 IgG. For primary tumor-bearing mice, clone-16-derived tumors responded well to anti-PD-L1 IgG therapy, as the tumor volume on the endpoint was significantly smaller when compared to the isotype control IgG therapy. In contrast, clone-1-derived tumors did not respond to the anti-PD-L1 IgG treatment. Although 4T1 parental-cell-derived tumors responded to anti-PD-L1 IgG therapy, the statistical difference between the control and anti-PD-L1 IgG group was smaller than in clone-16-derived tumor-bearing mice (Figure 2a). For evaluating therapeutic efficacy on liver metastasis, we measured the tumor diameter by imaging fluorescently labeled cancer cells utilizing IVM through a window chamber for 7 days. The size of the parental-cell-derived and clone-1-derived tumors did not show any significant difference, while the size of clone-16-derived tumors was found to have a more apparent difference between the two treatment groups (Figure 2b). In addition, tumor growth inhibition (Figure 2c) was determined based on the tumor size ratio of the endpoint to the initial point. The data revealed a difference in growth inhibition between two tumor sites, indicating a soil-dependent change in the therapeutic response of tumors to the same therapy. Overall, the heterogeneous response of tumors to anti-PD-L1 IgG immunotherapy can be dissected into the seed- (clone) and soil-dependent (TME) differences in anti-PD-L1 therapeutics efficacy.

### 3.3. IVM Imaging of Therapeutic Delivery and Efficacy on Liver Metastases

Seven days after the splenic injection of tumor cells, tumor-bearing mice received iv injection of anti-PD-L1 IgG conjugated with brilliant violet 421 (200 µg/kg). We then conducted studies utilizing IVM through a window chamber to determine the amount of delivered antibody in individual liver metastasis at 6 h after the injection. The mice were then also treated with anti-PD-L1 IgG therapy (15 mg/kg) on days 7 and 10. We monitored the individual tumor’s response to anti-PD-L1 IgG for 7 days and correlate the tumor size change with the amount of fluorescently labeled anti-PD-L1 IgG specifically delivered to each tumor (Figure 3). The tumor size changes were calculated based on the growth difference prior to and 7 days after therapy. In parental-cell-derived tumors, we found a significant inverse correlation between the amount of anti-PD-L1 IgG accumulation and its relative tumor size change (Figure 3a). Clone-1-derived tumors did not respond to anti-PD-L1 IgG, and the delivery of anti-PD-L1 IgG into tumors was limited (Figure 3b). The majority of clone-16-derived tumors disappeared after the therapy with a higher amount of anti-PD-L1 IgG accumulation than the clone-1-derived tumors (Figure 3b,c). These data indicated a significant clone-dependent difference in the amount of ani-PD-L1 IgG delivery to the tumor and its therapeutic efficacy. However, even when the same amount of therapy was delivered, therapeutic response varied among the tumors derived from all the cell lines. These data suggested that drug delivery cannot solely determine the therapeutic efficacy, and other factors may influence the results.

In this IVM imaging study, we also found that the fluorescently labeled anti-PD-L1 IgG accumulated preferentially at the border of the tumor (Figure 3). To explore the reason for the pattern of accumulation, we performed immunohistochemical staining of the liver section with the antibody against rat IgG to determine the delivery of iv injected rat anti-PD-L1 IgG as well as with antibody against mouse PD-L1 to evaluate PD-L1 expression. We reconfirmed that the delivery of rat anti-PD-L1 IgG into liver metastasis was limited at the border and did not reach the inside of the tumors. Meanwhile, the expression of PD-L1 fully covered the entire tumor (Appendix A). This result indicates that the delivery of therapeutic IgG is limited and cannot reach target cells efficiently, probably due to limited diffusion of the IgG in the TME.

### 3.4. Seed- and Soil-Dependent Difference in Tumor Coagulation

We then tried to determine the possible reason for the limited therapeutic delivery in the TME. Knowing that tumor coagulation can reduce the delivery of iv-injected therapy to tumor cells, we evaluated the amount of fibrinogen in the tumor sites by immunohistochemical staining [18,19]. Imaging quantification revealed that the fibrinogen area fraction was significantly higher in clone-1-derived tumors than both parental and clone-16-derived tumors, indicating a limited transport capability of antibody therapeutics in clone-1-derived tumors (Figure 4a,b). These data support the previous report of limited therapeutics delivery to tumor cells by coagulation inside the TME.

### 3.5. Immunohistochemical Analyses of Transport Properties for Anti-PD-L1 IgG in the Primary Tumor and Liver Metastases

In addition, we investigated other transport properties in tumors to determine the reasons for heterogeneous tumor response to anti-PD-L1 IgG in both primary tumor and liver metastases. First, we performed immunohistochemical staining of the tumor tissues using the antibody against Rat-IgG to quantify the amount of rat anti-PD-L1 IgG delivery to the tumor site 72 h after the treatment. Interestingly, both primary tumor and liver metastases showed an increased amount of anti-PD-L1 IgG accumulation in clone-16-derived tumors when compared to parental or clone-1-derived tumors (Figure 5a,b). Accumulation of anti-PD-L1 IgG was found not only at the border but also inside the tumor at 72 h after the injection, suggesting a time-dependent increase in the diffusion of therapy into the TME. Interestingly, the liver metastases of anti-PD-L1-treated mice showed a significantly greater amount of accumulation compared to the primary tumors originating from clone 16 cells. In the control set, the tumor-bearing mice were injected with rat isotype control IgG, and the IVM images did not show a specific accumulation of the IgG in tumors (data not shown). We also performed immunohistochemical analysis of nontreated tumor tissues to verify the PD-L1 expression in the TME. The result showed that the clone-16-derived tumors expressed a significantly higher amount of PD-L1 than parental or clone-1-derived tumors (Figure 5c,d). When comparing the PD-L1 expression in the clone-16-derived primary tumor and liver metastases, the data showed that liver metastases have a greater amount of PD-L1 expression. This finding further suggested the seed- (clone) and soil-dependent (TME) difference in PD-L1 expression and the delivery of anti-PD-L1 IgG to the TME. We then tried to identify cells that express PD-L1 in the TME of both primary tumor and liver metastases. Further analysis by double staining the sections using the antibody against PD-L1 and F4/80 or CD31 revealed that F4/80-positive macrophages and vascular endothelial cells partially expressed PD-L1 (Appendix A).

Next, we stained CD31 to determine whether the amount of vasculature presented in the tumors played an important role in the delivery of anti-PD-L1 IgG to the tumor site (Figure 5e). We did not find any significant difference in the amount of vasculature presented in the clones or between tumor sites (Figure 5f). This indicates that the amount of vasculature may not determine the difference in the delivery of therapy. However, the distinction in the TME, such as the high amount of fibrinogen in clone-1-derived tumors, may prevent blood flows in the vasculatures and thus alter the delivery of the therapeutics.

### 3.6. Immune Microenvironment

To determine the role of immune cells in the tumor microenvironment, the amounts of immune cells, including CD8, CD45, and F4/80, were imaged and quantified. Notably, a higher value of CD8, CD45, and F4/80 was found in the clone-16-derived tumors than the primary and clone-1-derived tumors, suggesting clone-16-derived tumors has a greater immune response (Figure 6). However, there is no apparent difference when comparing the amount of immune cells in between the primary tumor and liver metastases. These data suggest that the immune cell accumulation into the TME is more likely clone-dependent than organ-dependent.

Data from Table 1 summarize the immunohistochemical analyses we performed. The data suggested that heterogeneous tumor response to the anti-PLD1 Ab therapy could be attributed to (1) the difference in the expression of PD-L1 protein, (2) the variation in the amount of anti-PD-L1 IgG delivery to tumor cells, (3) the difference in the amount of tumor coagulation inside the tumor, and (4) the difference in the amount of immune cells inside the tumor.

## 4. Discussion

Variations in cancer cells (the “seed”), as well as interactions of the TME with cancer cells in various metastatic organs (the “soil”), can create complex inter- and intra-tumor heterogeneities [5,6]. These tumor heterogeneities can constitute a major source of therapeutic resistance against immunotherapy. For example, ICI using anti-PD-L1 IgG exhibited differential response rates at different organ sites, favoring melanoma patients with skin and lung metastases [20,21,22]. Lung cancer patients have shown shorter survival in the presence of liver metastases compared with other organ sites [23]. Understanding the specific driving forces in these tumor heterogeneities is crucial to yielding novel findings of therapeutic resistances, which may facilitate developing more effective personalized therapies.

It has also been reported that breast cancer metastases can occur approximately 51–63% in bone, 7–16% in the brain, 6–26% in the liver, and 17–30% in the lungs [24,25]. Even though the incidents of metastasis to other organs are higher than those to the liver, our study is focused on the liver due to the accessibility of the organ for IVM imaging. In this study, we used parental and single-cell-derived clonal populations of 4T1 murine breast cancer cells in producing both orthotopic primary and liver metastases models to deconvolute the complexity of polyclonal cancer cells and different TME-derived heterogeneities. One of the goals of our study is to correlate the amount of anti-PD-L1 IgG delivery to individual liver metastasis with its therapeutic efficacy. We incorporated IVM through a window chamber on live tumor-bearing mice to trace the amount of anti-PD-L1 IgG delivered to individual tumors and to correlate the therapeutic efficacy for seven days. The result indicates an inverse correlation between the amount of antibody therapy delivered to each tumor and the tumor growth inhibition in the mice bearing 4T1 parental-cell-derived tumors. We also found the PD-L1 expression level-dependent difference in anti-PD-L1 IgG delivery and therapeutic efficacy in clone-1- and clone-16-derived tumors: with clone 1 being low in PD-L1 expression and clone 16 having a higher one. These clone-dependent differences in PD-L1 expression and delivery of anti-PD-L1 IgG were changed by the location of tumors. Thus, our data are the first to report a seed- (clone) and soil-dependent (TME) difference in anti-PD-L1 therapeutic delivery and efficacy. However, we also found that when the amount of anti-PD-L1 IgG delivery was similar among the tumors regardless of the cell line, there was a difference in therapeutic efficacy. These variations indicate that, even when clonal cells are growing in the same TME, they do not always behave the same. Additional analysis at the single-cell level using the latest technologies will provide more information [26,27]. In addition, we will need to evaluate the temporal change in these heterogeneities, since when the tumors consisting of multiple clones are repeatedly treated with anti-PD-L1 IgG, only the clone-16-derived tumors disappear. In contrast, clone-1-derived tumors continue to survive and eventually become resistant to the therapy.

In the current clinical oncology, PD-L1 expression has been reported to be a valuable predictor of the efficacy of anti-PD-1/PD-L1 monotherapy in lung cancer patients [28]. In gastric cancer patients, PD-L1 expression in tumors is indicative of patient prognosis [29]. In melanoma patients, moderate PD-L1 expression has shown the best response rate by systemic anti-PD-1 therapy [30]. In these clinical studies, PD-L1 expression data were obtained only from needle biopsies and, therefore, cannot represent the entire tumor. Because of the expected inter- and intra-tumoral heterogeneity in the expression of PD-L1, these biopsy-based studies would have limitations in predicting the response of all the tumors to the therapy in the clinic.

To overcome these limitations, noninvasive nuclear imaging has been developed using radiolabeled anti-PD-L1 antibodies to evaluate PD-L1 expression in a preclinical setting. In these studies, SPECT/CT imaging was performed in xenograft tumor models using different cell lines, and the tumor sections were analyzed for PD-L1 expression by immunohistochemistry [31,32]. While SPECT/CT imaging showed a sufficient accumulation of the antibody in high PD-L1 expressing tumors, a limited uptake was observed in tumors with low or non-detectable levels of PD-L1 expression. We did not examine the pharmacokinetics of anti-PD-L1 antibodies, but the research indicates that 70% of anti-PD-L1 IgG was lost from circulation 7 days after injection in tumor-bearing mice with high anti-PD-L1 expression. Meanwhile, 6% of the antibody was lost in the circulation in tumor-bearing mice with low anti-PD-L1 expression [31]. These data suggest that iv injected radiolabeled antibodies could potentially be used for the noninvasive quantification of PD-L1 expression in tumors. These conclusions are based on the hypothesis that therapeutic antibodies in the systemic circulation can always reach target cells in the TME. Other studies imaging the distribution of iv-injected trastuzumab and cetuximab in HER2 and EGFR-positive tumors revealed a poor and heterogeneous distribution of therapeutics in the TME due to the limited convection and diffusion [33,34]. These data are in accord with our previous studies showing limited access of the circulating nanotherapeutics to tumor cells due to the various biophysical barriers in the TME [35,36,37]. Indeed, in our current study, the IVM images revealed a limited distribution of iv injected fluorescently labeled anti-PD-L1 IgG preferentially at the border, but not deep inside the tumor 6 h after the injection, while PD-L1 is expressed by the entire tumor. We have previously reported that the primary tumor and liver metastases of 4T1 breast cancer have limited blood perfusion and mass diffusivity inside tumors [16]. Therefore, the limited transport capability of the antibody therapeutics inside the tumor could create a gap between the area of cells expressing target protein and the amount of the therapeutic antibody delivered to the target cells. These gaps shall be considered for the interpretation of noninvasive SPECT/CT imaging data in humans. Negative imaging results of the tumor may not necessarily denote negative PD-L1 expression. The transport property in the TME shall be evaluated using a contrast reagent for CT imaging.

Histopathological analyses have revealed the presence of fibrin deposition and platelet aggregates inside and around different types of tumors, indicating an activation of the coagulation system by tumor cells, which can serve as a metastatic niche [38,39]. Leakage of coagulation factors from tumor-associated blood vessels into tumor interstitial spaces and the existence of tissue factors on tumor cells can induce local coagulation response not only inside the tumor blood vessels but also in the tumor stroma. Thus, a wide distribution of fibrin inside the TME can reduce blood flow in the vessels and compromise therapeutic delivery in tumors [18,19]. In our study, we identified a clone-dependent difference in the accumulation of fibrinogen inside tumors. The clone-1-derived tumors accumulated more fibrinogen when compared to parental-cell- and clone-16-derived tumors, while the accumulation of anti-PD-L1 was significantly lower in clone-1-derived tumors. These data support the previous studies that therapeutic delivery to tumor cells could also be hindered by coagulation. Indeed, treatment with a tissue plasminogen activator, a thrombolytic agent, could successfully deplete fibrin deposition, reopen the compressed tumor vessels, improve tumor blood flow, and further enhance the accumulation and penetration of nanotherapeutics [40].

Not only the PD-L1 expression levels in the TME but also the immune microenvironment has been shown to influence the prognosis of the disease and the immunotherapeutic efficacy. Activated T cell accumulation to the tumor is necessary for their anticancer cell function [41]. High-density invasive CD8+ in the TME cells are associated with prolonged survival in gastric cancer patients with ovarian metastases [42]. Interestingly, the PD-L1 expression level positively correlates with tumor-accumulating lymphocyte density in esophageal cancer [43]. Therefore, the efficacy of anti-PD-L1 immunotherapy can be predicted according to the degree of immune cell accumulation in the TME. In our study, there was a cell and organ type-dependent difference in immune cell infiltration into the TME, which was correlated with the effect of anti-PD-L1 IgG.

In addition to the PD-L1 expression, the tumor tolerance of therapy using anti-PD-L1 IgG is also an important factor that affects the efficacy. As a tumor-cell-extrinsic tolerance mechanism for the immunotherapy, various cell types, such as regulatory T cells, myeloid-derived suppressor cells, and tumor-associated macrophages present within the TME. These cells can express immunomodulatory factors, such as interleukin (IL)-4, -10, and -13, which suppress the antitumor immune response. Upon continuation of the current study, we will evaluate the seed- and soil-dependent difference in these tolerance mechanisms while increasing the number of clonal cell lines to be tested [44,45].

In conclusion, there was an inverse correlation between the amount of delivered anti-PD-L1 IgG and therapeutic efficacy in parental-cell-derived tumors. Tumors originating from clone 16 cells accumulated significantly more therapy and responded better than clone-1-derived tumors. The delivery of therapy and response were better when the clone-16-derived tumors resided in the liver compared to the primary site. A similar trend was found in PD-L1 expression and immune cell accumulation. In contrast, the amount of coagulation was significantly higher in clone-1-derived tumors compared to clone-16-derived tumors. Thus, we identified seed- and soil-dependent differences in PD-L1 expression, therapy delivery, immune cell accumulation, and tumor coagulation. All these differences in the TME can constitute heterogeneous delivery and immunotherapy response in polyclonal tumors growing in different organs. Further studies will be needed to overcome these heterogeneities to improve immunotherapeutic delivery and efficacy for cancer patients.

## Figures and Tables

**Figure 1 pharmaceutics-13-00530-f001:**
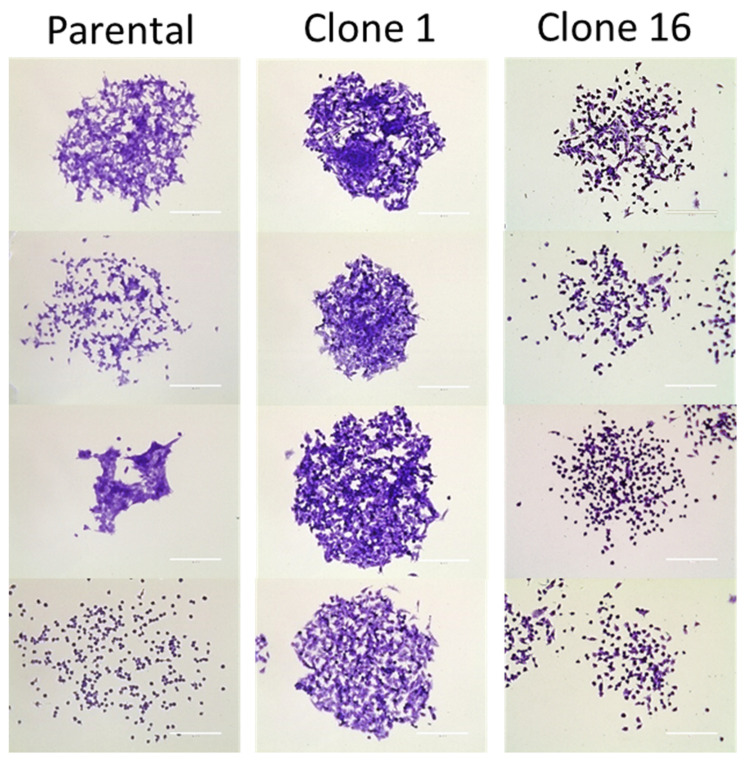
Clonogenic assay of 4T1 parental, clone 1, and clone 16 cells. In contrast, parental cells produced heterogeneous colonies, clone 1 and clone 16 cells produced clone-dependent similar colonies. Scale bar, 200 µm.

**Figure 2 pharmaceutics-13-00530-f002:**
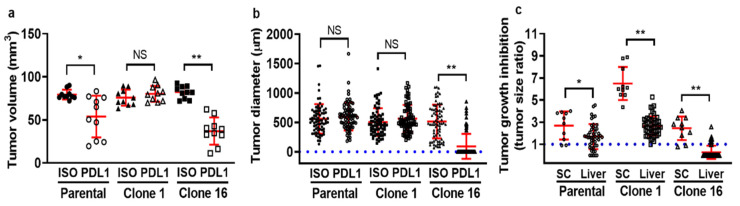
Determine seed- and soil-dependent differences in the therapeutic efficacy of tumor-bearing mice treated with either isotype control or anti-PD-L1 IgG. First, parental, clone 1, or clone 16 cells, were inoculated into the mammary fat pad (mfp) or spleen to produce primary tumor or experimental liver metastasis models. The tumor-bearing mice were then treated with either isotype control or anti-PD-L1 IgG on days 7 and 10. Primary tumor growth (**a**) and liver metastases growth (**b**) are shown. Tumor growth inhibition was determined by calculating the ratio of tumor volume or size between the endpoint to the initial time point (**c**). (* *p* < 0.05; ** *p* < 0.001 with respect to the isotype control IgG-treated mice and anti-PD-L1 IgG-treated mice; NS: no significant difference).

**Figure 3 pharmaceutics-13-00530-f003:**
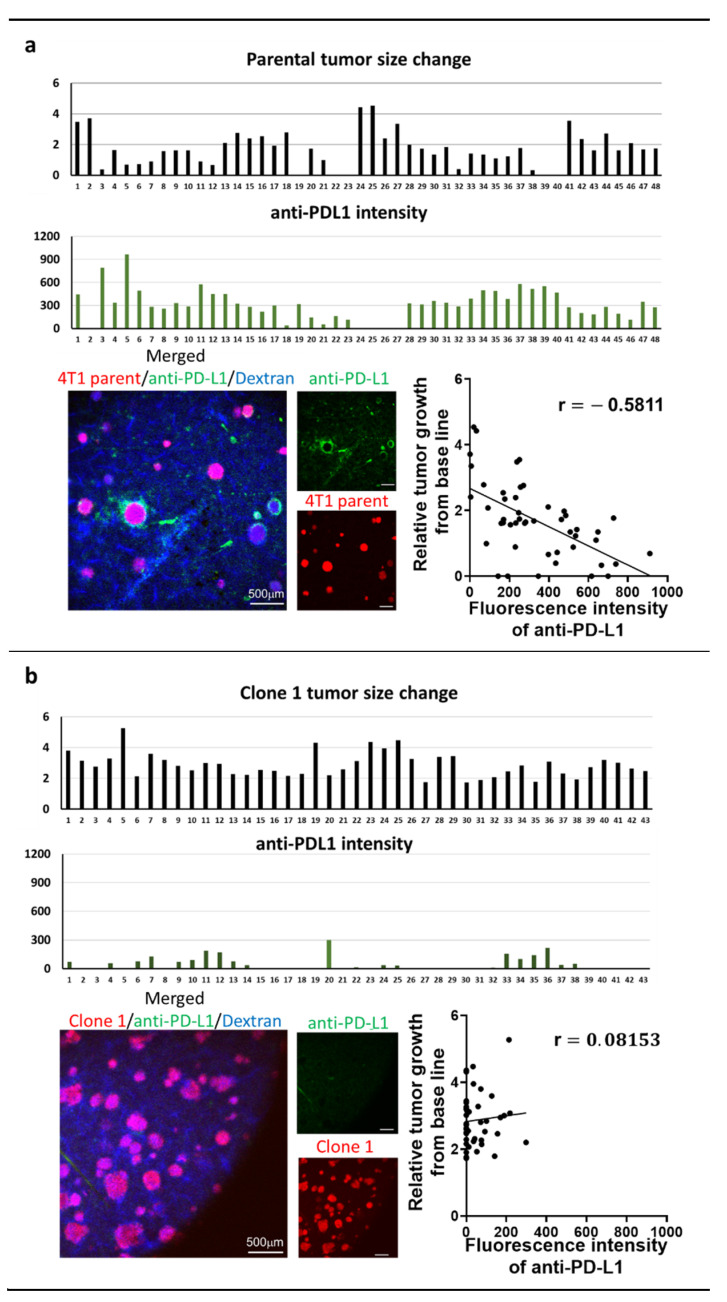
Intravital microscopy (IVM) imaging of liver metastasis of 4T1 parental (**a**), clone 1 or clone 16 cells treated with anti-PD-L1 IgG therapy through a window chamber for 7 days (**b**,**c**). IVM imaging of liver metastases originated by 4T1 parental, clone 1, or clone 16 cells (red) through a window chamber. Fluorescently labeled anti-PD-L1 IgG (200 µg/kg) was iv injected, and the fluorescence intensity of accumulated IgG in tumors was measured 6 h after injection. Mice were then further treated with non-labeled anti-PD-L1 IgG (15 mg/kg) and imaged for an additional 7 days. Tumor size change was calculated based on the measurement of tumor diameter prior to and seven days after therapy. Scattergrams showed the correlation between tumor size change and the amount of delivered anti-PD-L1 IgG in each tumor.

**Figure 4 pharmaceutics-13-00530-f004:**
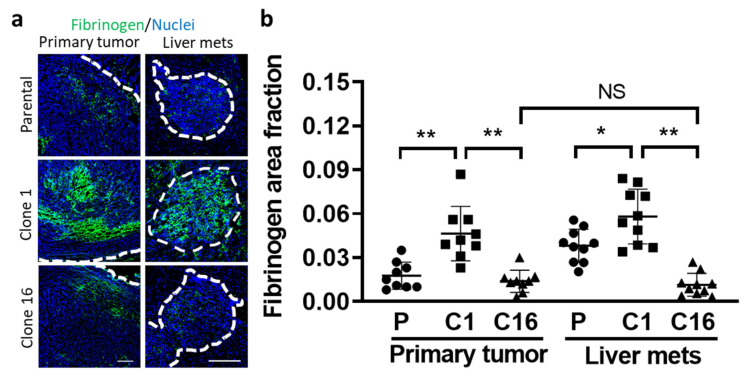
Tumor cell type and organ-dependent difference in coagulation. The amount of fibrinogen indicative of coagulation in tumors (surrounded by white dotted lines) was evaluated by immunohistochemical staining of the tissue sections and imaging using confocal microscopy (**a**). The area fraction of fibrinogen inside the tumors was compared between clones at two tumor sites (**b**). (* *p* < 0.05; ** *p* < 0.001; NS: no significant difference.) Scale bar, 100 µm.

**Figure 5 pharmaceutics-13-00530-f005:**
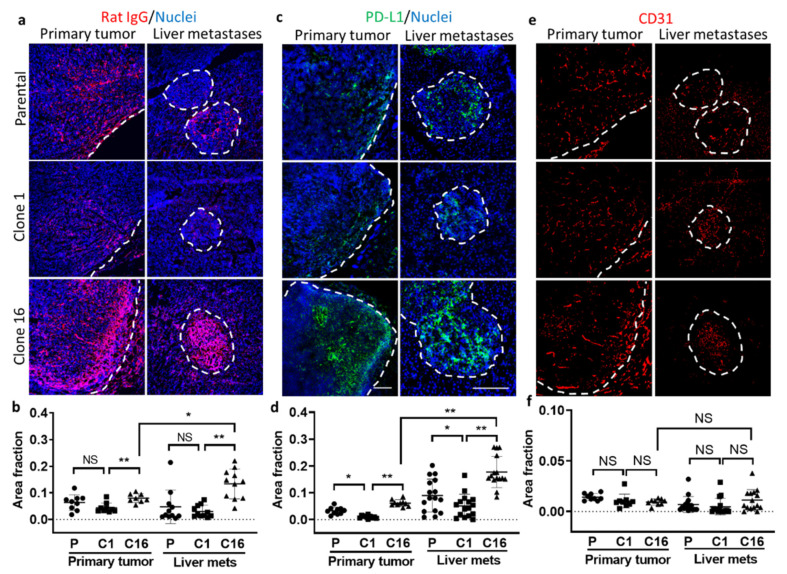
Immunohistochemical analyses of primary tumor and liver metastases. Immunofluorescence imaging (**a**) and quantification (**b**) of delivery of iv injected rat anti-PD-L1 IgG to tumors by immunostaining using the antibody against rat IgG. Immunofluorescence imaging (**c**) and quantification (**d**) of PD-L1 expression in parental-, clone-1-, clone-16-derived primary tumors and liver metastases using the antibody against PD-L1. Immunofluorescence imaging (**e**) and quantification (**f**) of vasculature in primary tumor and liver metastases using the antibody against CD31 (* *p* < 0.05; ** *p* < 0.001; NS: no significant difference). Scale bar, 100 µm.

**Figure 6 pharmaceutics-13-00530-f006:**
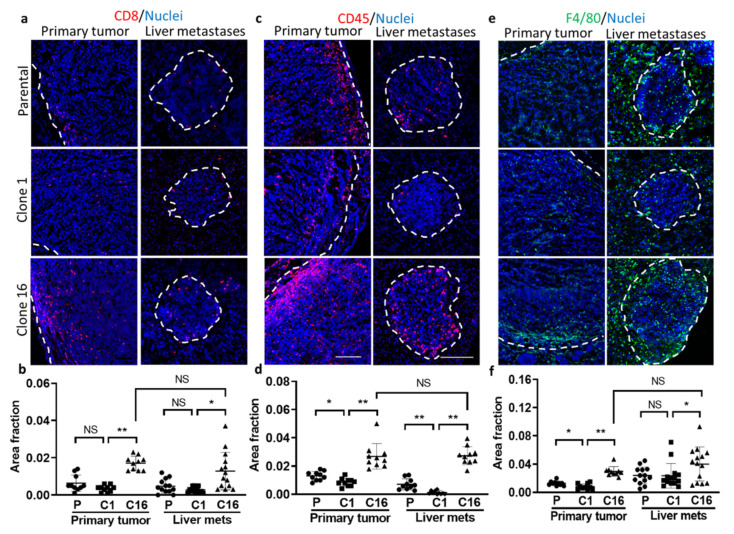
Evaluation of immune microenvironment in parental-, clone-1-, clone-16-derived primary tumor and liver metastases. Immunofluorescence imaging of (**a**) CD8, (**c**) CD45, and (**e**) F480 in both primary and liver metastases. Area fraction of (**b**) CD8, (**d**) CD45, and (**f**) F4/80 in parental-, clone-1- and clone-16-derived primary tumor and liver metastases are shown (* *p* < 0.05; ** *p* < 0.001; NS: no significant difference). Scale bar, 100 µm.

**Table 1 pharmaceutics-13-00530-t001:** Comparison of area fraction between Clone-1- and Clone-16-derived primary tumor and liver metastases.

	Delivery of anti-PD-L1		Expression of PD-L1		Fibrinogen		Vasculature		CD8		CD45		F4/80	
	Primary	Liver mets	Primary	Liver mets	Primary	Liver mets	Primary	Liver mets	Primary	Liver mets	Primary	Liver mets	Primary	Liver mets
**Clone 1**	0.0434	0.0302	0.0101	0.0511	0.0464	0.0581	0.0105	0.00481	0.00350	0.00277	0.00860	0.00128	0.00820	0.0236
**Clone 16**	0.0801	0.133	0.0615	0.177	0.0139	0.0115	0.0109	0.0112	0.0171	0.0129	0.0269	0.0274	0.0297	0.0401
***p***	<0.0001	<0.0001	<0.0001	<0.0001	<0.0002	<0.0001	NS	NS	<0.0001	<0.0015	<0.0001	<0.0001	<0.0001	<0.0015

## Data Availability

The data presented in this study are available on request from the corresponding authors.

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
