# Peer review of "Seed- and Soil-Dependent Differences in Murine Breast Tumor Microenvironments Dictate Anti-PD-L1 IgG Delivery and Therapeutic Efficacy"

_pharmaceutics, 2021, doi:10.3390/pharmaceutics13040530_

Round 1
Reviewer 1 Report
In this article, Kenji Yokoi et al. have a study on the seed and soil dependent differences in murine breast tumor microenvironments that affect the therapy of PD-L1 IgG. The study explained differences between the parental cell model and different single cell-derived clonal cell model including PD-L1 expression, immune cell accumulation and tumor coagulation. The study explained different therapeutic efficacy is caused by the heterogeneous tumor microenvironment, which will improve further studies of immunotherapeutic delivery. However, major revision is necessary before this article is accepted for publication on the “Pharmaceutics”. This decision is made based on some drawbacks listed as follows:
1) From line 160 to 179, this paragraph seems to be an analysis of Fig.2 instead of Fig.1.
2) Please add a scale bar for Fig.4.
3) The R2 in Fig.3 is too large. The date needs to be confirmed.
4) The metastases of breast tumor usually happen on the lungs, if the lung metastases have the same feature necessary to study.
5) The expression of PD-L1 is not the only factor to evaluate the response of immunotherapeutic, the tolerance of PD-L1 is also an important factor that affects therapeutic efficacy. The assay should be added to enrich the complication of the date.
Reviewer 2 Report
General comments:
The authors present an interesting investigation of how tumour sub-clones and the microenvironment determine the response of breast cancer cells to immune checkpoint inhibition. The study is well constructed, albeit it could have been strengthened by including additional breast cancer lines and sub-clones to ensure this was not an artifact of the specific cell line that was used. It could also have been strengthened by looking at interactions with cancer-associated fibroblasts at the primary and metastatic locations as they can play a major role in tumor fibrosis, vascularisation and immune infiltration (these points may be outside the scope of the current project and could be the subject of future studies). The manuscript does require a grammar check as there are errors throughout.
Specific comments:
- The flow of the introduction needs to be improved – it was unclear that breast cancer cells and immune checkpoint inhibitors were the focus of this study till the final paragraph.
- Line 84 has a full stop after “500”.
- Methods section 2.5 requires additional details: how were mice randomized? How was primary tumour size measured?
- Throughout the figures, it is unclear what the symbols refer to – were these multiple mice, multiple metastatic sites from one mouse/multiple fields of view from one mouse? Please clarify as this is critical to the interpretation of the reproducibility of the results.
- Results section 3.1, Figure 1: Was further characterisation of these cells performed in vitro? It would be good to know at at least a basic level what the phenotypic/genetic differences between the parent line and sub-clones is e.g. proliferation, expression of common surface markers, maintenance of parent line mutations?
- Figure 2C: “Growth inhibition” title on-axis is unclear, does this refer to the number of metastases?
- Line 256: “apparent amount of accumulation” – does this mean an increase? Please be more specific with the language used.
- Pages 8-9 appear to be missing in-text references to figures in results sections – please correct.
- Lines 279-280: it should also be specified that even though the amount of vasculature (CD31) was not different, there’s a possibility that the distortion of the vasculature may have been different if there were differences in fibrosis between the sites.
Reviewer 3 Report
Yokoi and coworkers reported the correlation between anti-PD-L1 IgG delivery and its therapeutic efficacy in tumor growth. The study is of interest for the in vivo applications of anti-PD-L1 delivery. The authors have evaluated the efficacy in tumors derived by two different clones. The work shall be accepted after addressing the following comments.
- The difference in PD-L1 expression may affect the capture of anti-PD-L1 antibodies. How much IgG loss was observed during circulation after 7 days and 10 days in terms of these two different clones?
- It would be great if the authors could include a table in the Discussion section to compare the difference between Clone 1 and Clone 16.
Round 2
Reviewer 1 Report
The authors have made enough revisions.